# Mammalian Arginase Inhibitory Activity of Methanolic Extracts and Isolated Compounds from *Cyperus* Species

**DOI:** 10.3390/molecules26061694

**Published:** 2021-03-18

**Authors:** Kamel Arraki, Perle Totoson, Alain Decendit, Andy Zedet, Justine Maroilley, Alain Badoc, Céline Demougeot, Corine Girard

**Affiliations:** 1PEPITE EA 4267, FHU INCREASE, University Bourgogne Franche-Comté, 25000 Besançon, France; arraki.kamel@yahoo.fr (K.A.); perle.totoson@univ-fcomte.fr (P.T.); andy.zedet@univ-fcomte.fr (A.Z.); justine.maroilley@edu.univ-fcomte.fr (J.M.); celine.demougeot@univ-fcomte.fr (C.D.); 2MIB-UR Oenologie, EA 4577, USC 1366 INRA, University of Bordeaux, ISVV, 33882 Villenave d’Ornon, France; alain.decendit@u-bordeaux.fr (A.D.); alain.badoc@u-bordeaux.fr (A.B.)

**Keywords:** *Cyperus thunbergii*, *Cyperus glomeratus*, arginase inhibitors, vasorelaxante activity, stilbenes, flavonoids

## Abstract

Polyphenolic enriched extracts from two species of *Cyperus*, *Cyperus glomeratus* and *Cyperus thunbergii*, possess mammalian arginase inhibitory capacities, with the percentage inhibition ranging from 80% to 95% at 100 µg/mL and 40% to 64% at 10 µg/mL. Phytochemical investigation of these species led to the isolation and identification of two new natural stilbene oligomers named thunbergin A-B (**1**–**2**), together with three other stilbenes, *trans*-resveratrol (**3**), *trans*-scirpusin A (**4**), *trans*-cyperusphenol A (**6**), and two flavonoids, aureusidin (**5**) and luteolin (**7**), which were isolated for the first time from *C.*
*thunbergii* and *C. glomeratus*. Structures were established on the basis of the spectroscopic data from MS and NMR experiments. The arginase inhibitory activity of compounds **1**–**7** was evaluated through an in vitro arginase inhibitory assay using purified liver bovine arginase. As a result, five compounds (**1**, **4**–**7**) showed significant inhibition of arginase, with IC_50_ values between 17.6 and 60.6 µM, in the range of those of the natural arginase inhibitor piceatannol (12.6 µM). In addition, methanolic extract from *Cyperus thunbergii* exhibited an endothelium and NO-dependent vasorelaxant effect on thoracic aortic rings from rats and improved endothelial dysfunction in an adjuvant-induced arthritis rat model.

## 1. Introduction

Arginase is a trimeric metalloenzyme hydrolyzing l-arginine to l-ornithine and urea. Arginase plays an important role in ammonia detoxification in mammals [1,2], but also in the regulation of the production of many key biological intermediates, such as polyamines (via l-ornithine), which are responsible for cell proliferation and collagen production (wound healing), and nitric oxide (NO) (via l-arginine availability), a strong vasodilating agent [1,3,4]. However, it is well known that excessive arginase activity can produce l-ornithine oversupply, which is responsible for tissue stiffening, and L-arginine depletion, which is responsible for a decrease in NO availability, by substrate competition with NO synthase [5]. This contribute to the appearance of endothelial dysfunction, which can be observed in various diseases for which an arginase overactivity has been implicated (inflammatory, cardiovascular, or cancer diseases) [6,7]. Previous studies conducted on animal models or in humans showed that inhibition of arginase enhanced NO bioavailability thereby restoring normal vascular function [8]. Even though this represents a new therapeutic strategy [9], no drug has been developed. Among the few synthetic arginase inhibitors commercially available, boronic acid derivatives (*S*-(2-boronoethyl)-l-cysteine (BEC), 2-(*S*)-amino-6-boronohexanoic acid (ABH), and Nω-hydroxy-nor-L-arginine (nor-NOHA) are the most potent, but their toxicity and poor pharmacokinetic profile limits their potential therapeutic use in humans [10]. Finding new arginase inhibitors suitable for the treatment of endothelial dysfunction associated with several diseases in humans still poses a challenge. Therefore, natural substances constitute a promising source in this area [11]. Among naturally occurring metabolites, flavonoids and stilbenoids have been shown to be the most active on arginase [11,12]. *Cyperus* genus attracted our attention because it possesses a wealth of these kinds of compounds.

The genus *Cyperus* belongs to the Cyperaceae family and includes more than 900 species distributed worldwide. *Cyperus* species constitute one of the three most widely and traditionally used genera in China [13], and *Cyperus rotundus* L., is prevalent in several systems of medicine (Ayurveda, Traditional Chinese Medicine, and medicine from Japan and Iran), where it is commonly used for treating a large variety of diseases (infectious and parasitic diseases, cancers, inflammation). Numerous pharmacological studies provide scientific evidence of the biological activities of these plants [14,15] because they possess an abundance of different phytochemicals. In general, *Cyperus* are renowned sources of biologically active compounds, such as essential oils [16], terpenes [17,18], coumarins, flavonoids [19,20,21], and stilbenes [21,22,23]. We recently isolated stilbene oligomers scirpusin B and cyperusphenol B from *Cyperus eragrostis* and demonstrated their in vitro inhibitory activity against liver bovine arginase [23]. In order to extend the research initiated on *Cyperus* genus, based on a survey of traditional uses and literature, we chose to continue with the study of the aerial parts of *Cyperus thunbergii* and *Cyperus glomeratus*. Arginase inhibitory activities of polyphenolic enriched extracts were investigated by means of in vitro and ex vivo studies. The isolation, identification, and arginase inhibitory activity of seven polyphenolic compounds, including two new stilbene oligomers, are also reported here.

## 2. Results and Discussion

### 2.1. In Vitro Arginase Inhibitory Activity of Cyperus thunbergii and C. glomeratus 

The aerial parts of two *Cyperus* species, *Cyperu*s *thunbergii* and *Cyperus glomeratus*, were extracted with methanol (MeOH) using maceration at room temperature. The dried extracts were subsequently subjected to solid phase extraction (SPE) in order to recover polyphenolic compounds. The in vitro evaluation [12] of these polyphenolic enriched extracts revealed that they were able to inhibit arginase (more than 80% inhibition at 100 µg/mL) (Figure 1).

### 2.2. Ex Vivo C. thunbergii Improved Endothelial Dysfunction in Arthritic Rats

The most active polyphenolic enriched extract, obtained from *C. thunbergii*, was evaluated for its effect on arginase-related endothelial dysfunction in the rat model of arthritis. At a severe stage of the arthritis model, endothelial dysfunction on the aorta vascular bed was attested by the depressed endothelium-dependent vasorelaxation in comparison to controls (Figure 2A). As expected [6], acetylcholine (Ach)-induced relaxation was higher in the presence of arginase inhibitor nor-NOHA (Figure 2B). Interestingly, *C. thunbergii* extract was also able to significantly improve this depressed relaxation. These data confirmed the arginase inhibitor activity of the present extract in ex vivo conditions. 

### 2.3. Isolation and Structural Elucidation of Compounds ***1***–***7***

In order to isolate active compounds, the phytochemical composition of both extracts was explored. This is the first phytochemical study of *C. thunbergii* and *C. glomeratus*. The reverse-phase preparative liquid chromatographic (PLC) investigation resulted in the isolation of two new stilbene compounds, thunbergin A (**1**) and thunbergin B (**2**), from *C. thunbergii*, together with five previously known compounds **3**–**7**, from *C. glomeratus* (Figure 3). 

The structures of **3**–**7** were established by comparing their observed data with those published in the literature, and identified as *trans*-resveratrol (**3**), *trans*-scirpusin A (**4**), aureusidin (**5**), *trans*-cyperusphenol A (**6**), and luteolin (**7**) (Table 1).

Compound **1** was obtained as a brownish powder amorphous solid. Its molecular formula, C_21_H_20_O_4,_ was deducted from high resolution electrospray ionization mass spectroscopy (HRESIMS), through the presence of a peak at *m/z* 337.1432 [M + H]^+^. All ^1^H, ^13^C, and distortionless enhancement by polarization transfer (DEPT) nuclear magnetic resonance assignments for **1** were performed using 2D NMR spectroscopic data: heteronuclear simple quantum correlation (HSQC) (Appendix A), heteronuclear multiple bond correlation (HMBC) (Appendix A), correlation spectroscopy (COSY) (Appendix A), and nuclear Overhauser effect spectroscopy (NOESY) (Appendix A). Twenty-one carbon signals were observed in the ^13^C NMR spectrum (Appendix A), discriminated by the DEPT experiment into two methyl, two methoxyl (OCH_3_) groups, six methane, and eleven quaternary carbon signals. Analysis of the ^1^H NMR spectrum (Appendix A) in methanol-*d*_6_ immediately revealed the presence of two groups of *ortho*-coupled aromatic protons at *δ*_H_ 6.76 (1H, d, *J* = 8.4 Hz) and *δ*_H_ 6.84 (1H, d, *J* = 8.4 Hz), four groups of *meta*-coupled aromatic protons *δ*_H_ 6.17 (1H, d, *J* = 2.2 Hz), *δ*_H_ 6.80 (1H, d, *J* = 2.2 Hz), *δ*_H_ 7.15 (1H, d, *J* = 2.2 Hz), and *δ*_H_ 6.61 (1H, d, *J* = 2.2 Hz), two methyls at *δ*_H_ 1.59 (6H, s), and two signals corresponding to methoxyl groups at *δ*_H_ 3.69 (3H, s) and *δ*_H_ 3.81 (3H, s). (Table 2). 

In the ^1^H–^1^H-COSY spectrum, correlations were observed between *δ*_H_ 6.76 d and *δ*_H_ 6.84 d (H-1/H-2), *δ*_H_ 6.61 d and *δ*_H_ 7.15 d (H-5/H-7), and *δ*_H_ 6.80 d and *δ*_H_ 6.17 d (H-9/H-11). The two methoxyl groups were placed to C-3 and C-10 as confirmed by HMBC correlations of the OCH_3_ (*δ*_H_ 3.81) with C-3 (*δ*_C_ 148.7) and the OCH_3_ (*δ*_H_ 3.69) with C-10 (*δ*_C_ 158.5) (Table 3). The location of the methyl groups was also concluded from the HMBC spectrum, as a proton signal at *δ*_H_ 1.59 (H_6_-13) showed correlations with *δ*_C_ 145.2 (C-4a), 130.2 (C-5a), and 36.9 (C-12). The hydroxyl groups linked to C-4 and C-6 were confirmed by the chemical shift of the quaternary carbons (*δ*_C-4_ 149.2 and *δ*_C-6_ 145.9). 

The NOESY correlations further confirmed the structure of compound **1**. Nuclear Overhauser effects were detected between H-2/H-1, H2/OCH_3_-3, H-5/H-13, H-9/H-7, H-9/OCH_3_-10, and H-11/OCH_3_-10 (Figure 4). All of the above confirmed the planar structure of compound **1** as 4,6-dihydroxy-12-dimethyl-3,10-dimethoxybenzophenanthrene, which was named thunbergin A (Figure 3).

Compound **2** was isolated as a brownish powder amorphous solid. It gave [M + H]^+^ at *m/z* 353.1386 in HRESIMS consistent with the molecular formula C_21_H_20_O_5_. All ^1^H (Appendix A), ^13^C (Appendix A), and J-modulated spin-echo (JMOD) NMR assignments for compound **2** were performed using 2D NMR spectroscopic data (HSQC (Appendix A), HMBC (Appendix A), COSY (Appendix A), NOESY (Appendix A)). Analysis of the ^1^H NMR spectrum in methanol-*d*_6_ (Table 4) displayed signals of two *meta*-coupled aromatic protons *δ*_H_ 6.63 (1H, d, *J* = 2.2 Hz, H-1) and *δ*_H_ 6.27 (1H, d, *J* = 2.2 Hz, H-3), two *ortho*-coupled aromatic protons *δ*_H_ 6.76 (1H, d, *J* = 8.4 Hz, H-8) and *δ*_H_ 6.65 (1H, d, *J* = 8.4 Hz, H-9), one aromatic proton as a singlet *δ*_H_ 6.66 (1H, s, H-5), two methyls, two methylenes, one sp^3^ methine, and signals of protons belonging to two methoxyls. In the JMOD spectrum, the presence of 21 carbon signals was detected (Table 3).

In the ^1^H–^1^H-COSY spectrum, correlations were observed between *δ*_H_ 6.63 d and *δ*_H_ 6.27 d (H-1/H-3), *δ*_H_ 6.76 d and *δ*_H_ 6.65 d (H-8/H-9), *δ*_H_ 2.73 dd and *δ*_H_ 3.08 m (H-11/H-12), and *δ*_H_ 2.96 dd and *δ*_H_ 3.08 m (H-11/H-12). The methine multiplet at *δ*_H_ 3.08, a methyl doublet at *δ*
_H_ 1.00, and two methylene protons at *δ*_H_ 2.73 dd and *δ*_H_ 2.96 dd provided evidence of the presence of a CH_3_CHCH_2_ structural unit (C-12, C-11) in the molecule. According to the ^1^H and ^13^C NMR signals at *δ*
_H_ 3.79 and *δ*
_C_ 54.4, and *δ*
_H_ 3.87 and *δ*_C_ 55.2, methoxyl groups could be identified and connected to C-2 and C-7, as confirmed by an HMBC correlation between OCH_3_ (*δ*_H_ 3.79) and C-2 (*δ*_C_ 160.5), and OCH_3_ (*δ*_H_ 3.87) and C-7 (*δ*_C_ 147.2) (Table 4). Moreover, on the basis of HMBC correlations between C-10/10-Me, C-11/10-Me, and C-5a/10-Me, the methyl groups (*δ*_H_ 1.79) were placed at C-10 (*δ*_C_ 38.0). On the basis of HMBC correlations between C-12/12-Me, C-11/12-Me, and C-10/12-Me, the methyl groups (*δ*_H_ 1.00) were placed at C-12 (*δ*_C_ 50.2). The NOESY correlations further confirmed the structure of compound **2**. Nuclear Overhauser effects were detected between H-1/OCH_3_-2, H3/OCH_3_-2, H-5/H-11, H-5/C, H_3_-10, and H-8/OCH_3_-7 (Figure 4), which was named thunbergin B (Figure 3).

Table 1 shows the chromatographic data set and the *m/z* values of the isolated compounds. NMR and HRESIMS data were used to identify compounds **5** and **7** as new compounds from the *Cyperus* genus. 

Aureusidin (**5**) belongs to the less studied subclass of flavonoids called aurones. This molecule, which rarely occurs in nature, was previously isolated from mosses marine brown algae and flowering plants [24,25]. Aureusidin possesses a pharmacological profile showing high antioxidant and lipoxygenase inhibitory activity [26], as well as anti-inflammatory effects [27,28]. Luteolin (**7**), a flavone, which is abundant in edible plants, displays a wide range of biological activities including antioxidant, anti-carcinogenic [29], cardioprotective [30], anti-inflammatory, and antipruritic [31] activities. Scirpusin A (**4**) and cyperusphenol A (**6**) are stilbene oligomers, most of which are potent antioxidants showing cardioprotective properties. Their structures result from the condensation of resveratrol and piceatannol. Scirpusin A (**4**), a hydroxystilbene dimer, acts as an effective singlet oxygen quencher and DNA damage protector [32].

### 2.4. Arginase Inhibitory Activity of Compounds ***1***–***7***

Compounds **1**–**7** were screened for their arginase inhibitory activity using purified bovine liver arginase [12]. Their IC_50_ values are indicated in Table 5. 

The synthesized compound Nω-hydroxy-nor-l-arginine (nor-NOHA), a well-known reference inhibitor of arginase, was used as a positive control (IC_50_ = 1.7 ± 0.7 µM). Although all of the evaluated compounds remained less active than nor-NOHA, it should be noted that compound **4** (IC_50_ = 17.6 ± 2.2 µM), compound **6** (IC_50_ = 19.4 ± 1.3 µM), and compound **1** (IC_50_ = 28.8 ± 2.5 µM) all show an activity close to that of the natural inhibitor piceatannol (IC_50_ = 12.6 ± 0.6 µM), one of the most active natural compounds on mammalian arginase [12,23]. 

## 3. Materials and Methods

### 3.1. Reagents

All reagents were from Sigma-Aldrich (Saint-Quentin Fallavier, France). They were used without further purification, except for purified liver bovine arginase 1, which was purchased from MP Biomedicals (Illkirch-Graffenstaden, France) (one unit (1U) of bovine arginase corresponds to the amount of enzyme able to convert 1 µMol of L-arginine to urea and L-ornithine per minute at pH 9.5 and 37 °C). MeOH, acetonitrile (MeCN), and dimethylsulfoxide (DMSO) were obtained from two companies: Carlo Erba Reagents (Val de Reuil, France) and VWR Chemicals (Fontenay-sous-Bois, France). Deuterated solvents and trifluoroacetic acid (TFA) were purchased from Eurisotop (Tewksbury, MA, USA) and Fisher Scientific (Illkirch, France), respectively. Water was purified (resistivity > 18 mΏ/cm) using an water purification system (ELGA LabWater, UK).

### 3.2. Plant Materials

*Cyperus thunbergii* Vahl. (XX-0-TUEB-3630 ex JB Tubingen) and *Cyperus glomeratus* (FR-0-LYJB-005964W ex JB Lyon) aerial parts were collected in the Botanical Garden of Talence (Talence, France) between 2017 and 2018. Each plant was authenticated by one of the authors (A.B). The samples were thoroughly dried and kept free from moisture.

### 3.3. Extraction and Isolation

The aerial parts of *C. thunbergii* and *C. glomeratus* were ground into powder. A sample of each (80 g) was extracted, then macerated and stirred in methanol at room temperature (600 mL × 5 × 24 h). The methanolic solutions were recovered by filtration, then pooled and concentrated under reduced pressure to obtain dry extracts. These crude extracts (12 g) were dissolved in 30% MeOH (1 g of extract in 600 µL of MeOH and 1.4 mL of H_2_O) by vortexing and sonicating. Each extract was pre-purified using a solid phase extraction (SPE) mini column Strata^®^ C_18_-E (55 µM, 70Å). Each sample (2 mL) was loaded onto the C_18_ mini column, washed with H_2_O 4 mL water, and then eluted with 90% MeOH. The recovered solution contained polyphenols. Each extract was evaporated until dry, using the same vacuum evaporator. Before HPLC analyses, the dried extract was redissolved in 50% MeOH HPLC grade by vortexing and sonicating, before filtration through an Acrodisc^®^ (25 mm Syringe Filters) 0.2 μm nylon HPLC-certified membrane.

### 3.4. Identification of Pure Compounds

Identification and structural elucidation of the purified compounds were carried out on a mass spectrometer (high-resolution electrospray ionization mass spectra or HRESIMS) and an NMR spectrometer. HRESIMS data were acquired on an SCA Illkirch QToF instrument. ^1^H NMR at 400 MHz and ^13^C NMR data at 100 MHz were acquired using a Bruker AC300 spectrometer (Bruker BioSpin, Billerica, MA, USA). All compounds were dissolved in methanol-*d*_4_ and acetone-*d*_6_ for 1D NMR and 2D NMR measurements (including COSY, HSQC, NOESY, and HMBC). Chemical shifts (*δ*) were reported in parts per million (ppm) relative to the residual solvent signals. Coupling constants (*J*) were reported in Hz. Data were presented as follows: chemical shift (*δ*, ppm), multiplicity (s, singlet; br s, broad singlet; d, doublet; dd, doublet of doublets; t, triplet; q, quartet; m, multiplet), coupling constant (*J*, Hz), integration.

The purification was achieved through preparative liquid chromatography (PLC). Polyphenolic extracts were separated on a Gilson PLC 2020 Kinetex^®^ EVO reverse-phase C_18_ column (250 × 21.2 mm, 5 µM). The solvent system used was ultrapure H_2_O acidified with 0.1% TFA (solvent A), and MeCN acidified with 0.1% TFA (solvent B). The elution program at 20 mL/min was 20% B (0–5 min), 20–60% B (5–35 min), 60% B (35−45 min), followed by a 5 min wash with 100% B. The injections were 500 µL with a concentration of 50 mg/mL. The chromatograms were registered at 286 and 306 nm. Preparative PLC performed on *Cyperus thunberghii* extract yielded two novel compounds: **1** (4.3 mg, *t*_R_ 19.7 min) and **2** (12.7 mg, *t*_R_ 21.5 min), whereas PLC performed on *C. glomeratus* yielded compounds **3** (4.6 mg, *t*_R_ 13.8 min), **4** (15.1 mg, *t*_R_ 18.5 min), **5** (3 mg, *t*_R_ 29.2 min), **6** (6.3 mg, *t*_R_ 22.5 min), and **7** (4.8 mg, *t*_R_ 36.6 min). Compound purity was controlled by analytical HPLC. Compounds **3**–**7** were respectively identified as resveratrol [33], *trans*-scirpusin A [32], aureusidin [26], *trans*-cyperusphenol A [22], and luteolin [34] through a comparison with the data reported in the literature.

### 3.5. Measurement of Arginase Activity

#### 3.5.1. In Vitro with Bovine Arginase

The amount of urea produced by the hydrolysis of L-arginine by arginase (purified liver bovine arginase (b-ARGI)) can be detected using a color reactant (α-isonitrosopropiophenone) followed by a colorimetric assay, as described below. In each well of a 96-well microplate, the solutions were added in the following order: (1) buffer containing Tris-HCl (50 mM, pH 7.5) and 0.1% of bovine serum albumin (TBSA buffer) (10 μL), with or without (control) arginase (0.025 U/μL); (2) Tris-HCl solution (50 mM, pH 7.5) containing 10 mM MnCl2 as a cofactor (30 μL); (3) a solution containing an inhibitor or its solvent (as a control) (10 μL); (4) a solution of L-arginine (pH 9.7, 0.05 M) (20 μL). The microplate was incubated for 60 min in a 37 °C water bath after covering with a plastic sealing film. The addition of 120 μL of H_2_SO_4_/H_3_PO_4_/H_2_O (1:3:7) quenched the reaction. The microplate was left on ice for 5 min. Thereafter, 10 μL of α-isonitrosopropiophenone (5% in absolute ethanol (EtOH)) was added, and the microplate was heated in an oven at 100 °C for 45 min, after covering with an aluminum sealing film. As the colored product is photosensitive, the microplate was kept in the dark until reading. After 5 min of centrifugation and cooling for another 10 min, the microplate was shaken for 2 min and the absorbance was read at 550 nm and 25 °C using a spectrophotometer (Synergy HT BioTeck). The level of arginase activity was expressed as relative to the “100% arginase activity”. The experiment was repeated three times with each microplate under similar experimental conditions (e.g., various inhibitor concentrations.)

The percentage of arginase inhibitory activity and IC_50_ values was evaluated as previously described [12]. A stock solution (70 mM) was prepared in DMSO and stored at −26 °C for each compound. These stock solutions were extemporaneously and successively diluted in ultrapure H2O to afford the following concentrations: 7000, 2100, 700, 210, 70, 21, 7, 2.1, and 0.7 μM, corresponding to final concentrations in the wells of 1000, 300, 100, 30, 10, 3, 1, 0.3, 0.1 μM, respectively. For a first screening, compounds were tested at final concentrations of 10 and 100 μM. Each solution was incubated with arginase for 1 h, as described above. The percentage of arginase inhibition was calculated by conversion of the resulting absorbance (relative to the absorbance of controls with only solvent (“100% arginase activity”)) and plotted on a semilogarithmic scale. The IC_50_ values were estimated by nonlinear sigmoidal curve-fitting by using Prism (GraphPad Software, version 5.0.3).

#### 3.5.2. Ex Vivo, in Isolated Aortic Rings from Arthritic Rats

This part of the experiment was performed on 15 male Lewis rats (6 weeks old), purchased from JanvierLabs (Le Genest Saint Isle, France). The experimental procedures were approved by the local ethics committee for animal experimentation No. 2015/001-CD/5PR of Franche-Comté University (Besançon, France) and complied with the “Animal Research: Reporting In Vivo Experiments” (ARRIVE) guidelines. 

Arthritis was induced by a single intradermal injection to the tail of 120 μL of 10 mg·ml^−1^ heat-killed *Mycobacterium butyricum* suspended in Freund’s incomplete adjuvant, as described previously [35]. Non-arthritic age-matched rats were used as controls and received saline at the base of the tail. 

At 33 days post-immunization, corresponding to the acute phase of arthritis, the rats were anesthetized using sodium pentobarbital 60 mg/kg (Ceva Santé Animale, France). The descending thoracic aorta was excised and carefully cleaned for vascular study as previously described [23]. Acetylcholine (10^−11^−10^−4^ M) relaxation curves were achieved in aortic rings from adjuvant induced-arthritis and controls rats. To assess the effect of arginase inhibitors, experiments were repeated in the presence of nor-NOHA (10^−4^ M) and *C. thunbergii* extract (at EC_50_ obtained by in vitro test), respectively.

### 3.6. Data and Statistical Analysis

Values were presented as means ± SD. Data were analyzed with Prism (GraphPad Software, version 5.0.3). The comparison between two values was assessed by unpaired Student’s *t* test or Mann–Whitney U test when data were not normally distributed. Concentration-response curves were compared by two-way analysis of variance (ANOVA) for repeated measures. A *p* < 0.05 was considered significant.

## 4. Conclusions

In conclusion, studies were carried out on polyphenolic enriched methanolic extracts from aerial parts of *Cyperus thunbergii* and *C. glomeratus*, due to their interesting mammalian arginase inhibitory effect. Seven compounds were isolated for the first time from these two species, two of which are new stilbenes: thunbergin A (**1**) and B (**2**). Compounds **1**, **4**–**7** showed arginase inhibitory activities close to those of the natural reference inhibitor piceatannol. Firstly, our results suggest that polyphenolic enriched extracts from *Cyperus* species constitute a valuable source from which to discover new natural arginase inhibitors. Notably, *C. thunbergii* extract improved endothelial dysfunction in arthritic rats. Secondly, these data highlight the potential benefits of polyphenolic-enriched extracts or stilbenes-type compounds isolated from *Cyperus* sp. for the vascular management of arthritis via an arginase inhibitory activity.

## Figures and Tables

**Figure 1 molecules-26-01694-f001:**
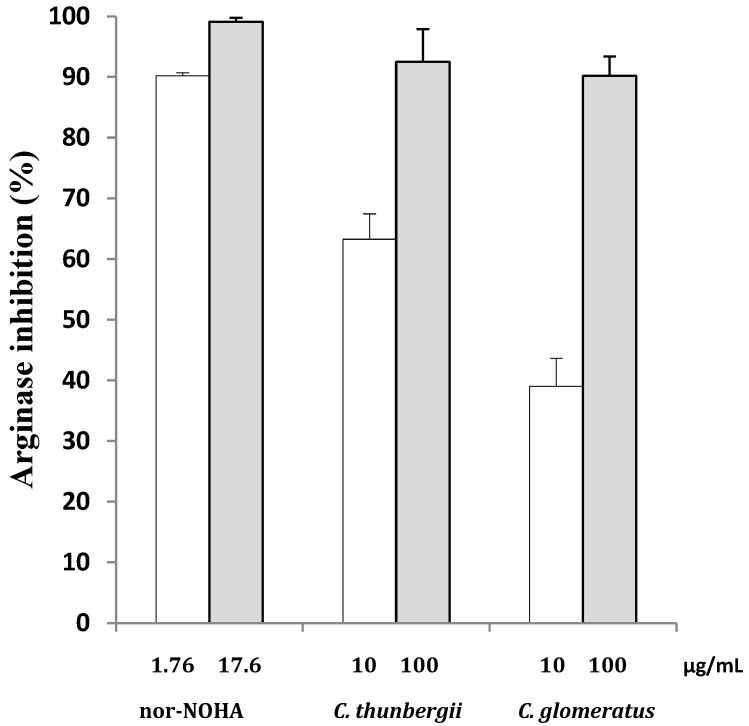
Arginase inhibition (at 10 and 100 μg/mL) of the methanolic extracts of *Cyperus*
*thunbergii*, *Cyperus glomeratus,* and Nω-hydroxy-nor-l-arginine (nor-NOHA) (positive control at 10 and 100 μM corresponding to 1.76 and 17.6 μg/mL, respectively). Results are expressed as means ± standard deviation (SD) obtained from three distinct experiments performed in duplicate.

**Figure 2 molecules-26-01694-f002:**
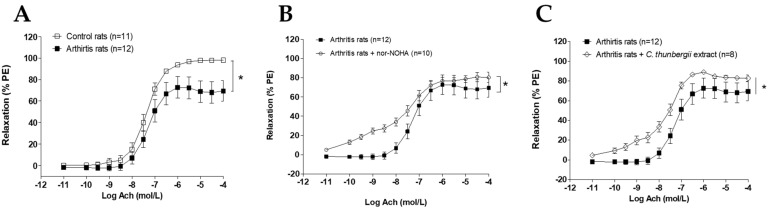
Acetylcholine (Ach)-induced relaxation in arthritic rats and the effect of *Cyperus thunbergii* extract. Experiments were performed on aortic rings from control and arthritis rats on day 33 post-immunization. Arteries were constricted with phenylephrine (10^−6^ mol/L) and relaxed with cumulative concentrations of Ach (**A**). The same experiments were performed in rings from arthritis rats in the presence of nor-NOHA (10^−4^ mol/L) (**B**) and *Cyperus thunbergii* extract (2.6 10^−3^ mg/mL) (**C**). Values are means ± standard error of mean (SEM) from *n* = number of aorta rings. * (*p* < 0.05).

**Figure 3 molecules-26-01694-f003:**
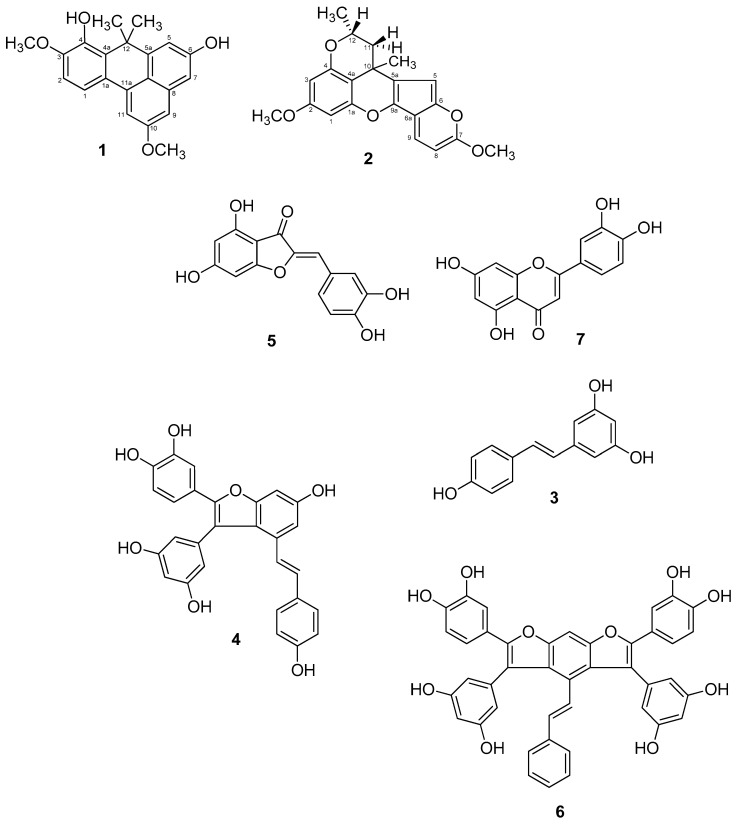
Structures of compounds **1**–**7**.

**Figure 4 molecules-26-01694-f004:**
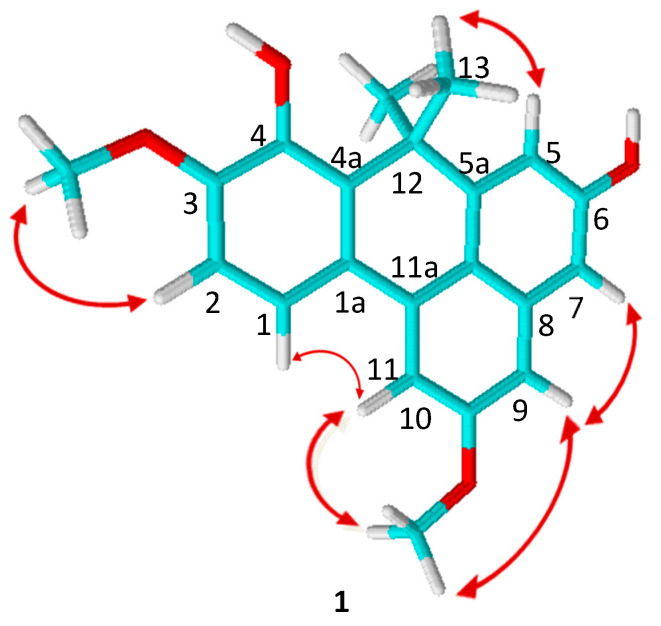
Key NOESY correlations for compounds **1** and **2**.

**Table 1 molecules-26-01694-t001:** Identification of compounds using retention times and electrospray ionization mass spectroscopy (ESIMS) data.

Compound	*t*_R_ (min)	[M + H]^+^	Identification
*Cyperus thunbergii*			
**1**	19.7	337	compound **1**
**2**	21.5	353	compound **2**
*Cyperus glomeratus*			
**3**	13.8	229	resveratrol
**4**	18.5	471	*trans*-scirpusin A
**5**	29.2	287	aureusidin
**6**	22.5	713	*trans*-cyperusphenol A
**7**	36.6	287	luteolin

**Table 2 molecules-26-01694-t002:** NMR spectroscopic data for compound **1** (400 MHz, Methanol-*d*_4_).

Position	*δ_H_*(m, *J* in Hz)	*δ*_C_, Type	COSY (H→H)
1	6.84 (1 H, d, *J* = 8.4)	121.6, CH	H_2_
1a		126.5, C	
2	6.76 (1 H, d, *J* = 8.4)	108.3, CH	H_1_
3		148.7, C	
4		149.2, C	
4a		145.2, C	
5	6.61 (1 H, d, *J* = 2.2)	117.9, CH	H_7_
5a		130.2, C	
6		145.9, C	
7	7.15 (1 H, d, *J* = 2.2)	125.3, CH	H_5_
8		135.3, C	
8a		137.0, C	
9	6.80 (1 H, d, *J* = 2.2)	98.3, CH	H_11_
10		158.5, C	
11	6.17 (1 H, d, *J* = 2.2)	100.3, CH	H_9_
11a		123.0, C	
12		36.9, C	
13	1.59 (6 H, s)	28.5, CH_3_	
3-OMe	3.81 (3 H, s)	55.1, CH_3_	
10-OMe	3.69 (3 H, s)	54.6, CH_3_	

The coupling constants (*J*) are given in parentheses and reported in Hz; chemical shifts (*δ*) are given in ppm.

**Table 3 molecules-26-01694-t003:** Major HMBC correlations for compounds **1** and **2**.

	1	2	
Position	HMBC (H→C)	Position	HMBC (H→C)
1	2, 3, 1a, 4a	1	3, 4a, 2, 1a
2	1, 3, 1a	3	1, 4a, 4, 2
5	8a, 6, 7, 12, 13	5	5a, 10, 6
7	8a, 5, 6, 8	8	9, 6, 6a, 7
9	7, 8, 10, 11	9	8, 7, 9a, 6a
11	11a, 9, 10	11	10, 12, 4a, 1a, 4
13	4a, 5a, 12	12	12-Me, 11, 10
3-OMe	3	10-Me	10, 11, 5a
10-OMe	10	12-Me	12, 11, 10
		2-OMe	2
		7-OMe	7

**Table 4 molecules-26-01694-t004:** NMR spectroscopic data for compound **2** (400 MHz, Methanol-*d*_4_).

Position	*δ_H_* (m, *J* in Hz)	*δ*_C_, Type	COSY (H→H)
1	6.63 (1 H, d, *J* = 2.2)	95.6, CH	H_3_
1a		142.2, C	
2		160.5, C	
3	6.27 (1 H, d, *J* = 2.2)	101.2, CH	H_1_
4		154.1, C	
4a		124.6, C	
5	6.66 (1 H, s)	115.5, CH	
5a		128.9, C	
6		144.6, C	
6a		129.2, C	
7		147.2, C	
8	6.76 (1 H, d, *J* = 8.4)	108.1, CH	H_9_
9	6.65 (1 H, d, *J* = 8.4)	118.1, CH	H_8_
9a		142.1, C	
10		38.0, C	
11	2.73 (1 H, dd, *J* = 16.2, 6.9) 2.96 (1 H, dd, *J* = 16.2, 9.1)	27.3, CH	H_12_
12	3.08 (1 H, m)	50.2, CH	H_11_
10-Me	1.79 (3 H, s)	26.2, CH_3_	
12-Me	1.00 (3 H, s)	17.9, CH_3_	
2-OMe	3.79 (3 H, s)	54.4, CH_3_	
7-OMe	3.87 (3 H, s)	55.2, CH_3_	

The coupling constants (*J*) are given in parentheses and reported in Hz; chemical shifts (*δ*) are given in ppm.

**Table 5 molecules-26-01694-t005:** Arginase inhibitory activity of compounds **1**–**10**
*^a^*.

Compound	Arginase Inhibition, IC_50_ (µM)
nor-NOHA *^b^*	1.7 ± 0.2
**1**	28.8 ± 2.5
**2**	74.1 ± 3.7
**3**	105.2 ± 4.1
**4**	17.6 ± 2.2
**5**	57.1 ± 2.3
**6**	19.4 ± 1.3
**7**	60.6 ± 3.1

*^a^* Values are means ± SEM and were obtained from three distinct experiments performed in triplicate. *^b^*
*p* < 0.05, significantly different from nor-NOHA (reference inhibitor).

## Data Availability

The data presented in this study are available on request from the corresponding author.

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
