# Peer review of "Mammalian Arginase Inhibitory Activity of Methanolic Extracts and Isolated Compounds from Cyperus Species"

_molecules, 2021, doi:10.3390/molecules26061694_

Round 1

Author Response

Response to Reviewer 1

 Response:

We thank the reviewer for his/her comments that improved the quality of our manuscript.

All the changes made in the manuscript are indicated in yellow in the text.

- As requested, english language has been improved by revision by a native English speaker. The modifications are highlighted in green in the text.

- We agree with the reviewer, a list of abbreviations would be usefull. However, the instructions to authors states thatAbbreviations should be defined in parentheses the first time they appear in the abstract, main text, and in figure or table captions and used consistently thereafter.” So abbreviations in the text and figure legends were revised.  We still added a list of abbreviations on the first page of the manuscript.

- We have made all the changes proposed and we have corrected the mistakes noticed by the reviewer, in particular:

  • conglomeratus” has been replaced by “glomeratus” in the text,
  • The term “appealing” has been removed.
  • “their versatile structures” has been replaced by “their structures”

- We also thank the reviewer for the suggestion of more references about the activity of aureusidin. These new references (ref 27 and 28) have now added in the revised manuscript.

- The reviewer is right : the only literature data about the compound piceatannol as reference inhibitor are self references, so we choose to remove the term « reference », in the abstract, in the section 2.3 and in the conclusion: « the natural arginase reference inhibitor, piceatannol» has been replaced by « the natural arginase inhibitor, piceatannol»

- We agree with the reviewer’s comment. Herein, we reported that Cyperus thunbergii extract exhibited vasorelaxant activities by activating endothelial NO synthase. However, this vasorelaxant effect was not directly connected to its arginase inhibitory property. So as requested, we focus the aim and the discussion of our findings on the arginase inhibitory activities of pure compounds.

Vasorelaxant part was suppressed and the title has been amended accordingly. However, the effect of the extract on endothelial dysfunction associated to rat model of arthritis was maintained, as this result demonstrated the arginase inhibitor effect of the extract in ex vivo condition.

We change the objectives as follows:

Arginase inhibitory activities of polyphenolic enriched extracts were investigated by means of in vitro and ex vivo studies. The isolation, identification and arginase inhibitory activity of seven polyphenolic compounds including two new stilbene oligomers, were also reported here.”

instead of

“Arginase inhibitory activity and vasorelaxant properties of polyphenolic enriched extracts were investigated, as well as their effects on endothelial function and signalling pathways. Isolation, identification and arginase inhibitory activity of seven polyphenolic compounds including to new stilbene oligomers, were also reported here.”

- Figure 2 was suppressed, Fig 3 became Figure 2 revised.

- Conclusion paragraph was also updated as following:

In conclusion, studies were carried out on polyphenolic enriched methanolic extracts from aerial parts of Cyperus thunbergii and C. glomeratus, due to their interesting mammalian arginase inhibitory effect. Seven compounds were isolated for the first time from these two species, two of which are new stilbenes: thunbergin A (1) and B (2). Compounds 1, 4-7 showed arginase inhibitory activity close to those of the natural reference inhibitor, piceatannol. Firstly, our results suggest that polyphenolic enriched extracts from Cyperus species constitute a valuable source with which to discover new natural arginase inhibitors. Notably, C. thunbergii extract improved endothelial dysfunction in arthritic rats. Secondly, this data highlights the potential benefits of polyphenolic enriched extracts or stilbenes-type compounds isolated from Cyperus sp. for the vascular management of arthritis, via an arginase inhibitory activity.”

- According to the reviewer’s comment, we improved the manuscript plan, as following:

  1. Introduction
  2. Results and discussion

2-1 Arginase inhibitory activity of Cyperus thunbergii and C. glomeratus

      In vitro

      In vivo

2.2. Isolation and structural elucidation of compounds 1-7

2-3 Arginase inhibitory activity of compounds 1-7

  1. Materials and methods

3-1 Reagents

3-2 Plant Materials

3-3 Extraction and Isolation

3-4 Identification of pure compounds

3-5 Measurement of arginase activity

3-6 Data and Statistical Analysis

- All the minor changes and the corrections proposed by the reviewer have been made in the revised manuscript.

****

We hope that the revised manuscript comes up to reviewer’s expectations and that the reviewer will consider the corrected version acceptable for publication in Molecules.

Reviewer 2 Report

The ms: Mammalian Arginase Inhibitory Activity and Vasorelaxant Positive Effect on Thoracic Aorta Rings from Rats of Methanolic Extracts from Cyperus Species, contains the description of the chemical composition of two Cyperus species and two of the isolated compounds are new. Also, describe the correlation with its biological activity in the selected model. The reading of the ms is easy to follow, but there are some details that dismiss its quality:

i. page 1 and 2, lines 37, 55, and 57 why use ..... instead to complete the information?

ii. In line 66 page 2 says about signaling pathways, but in results, discussion or conclusions nothing is said about that.

iii In line 389, page 14 one of the units has a rare symbol, what is this?

Author Response

Response to Reviewer 2

Comments of reviewer 2.

The ms: Mammalian Arginase Inhibitory Activity and Vasorelaxant Positive Effect on Thoracic Aorta Rings from Rats of Methanolic Extracts from Cyperus Species, contains the description of the chemical composition of two Cyperus species and two of the isolated compounds are new. Also, describe the correlation with its biological activity in the selected model. The reading of the ms is easy to follow, but there are some details that dismiss its quality:

  1. page 1 and 2, lines 37, 55, and 57 why use ..... instead to complete the information?

According to the reviewer’s comment, we have added all complete information and suppressed the “…” symbols.

  1. In line 66 page 2 says about signaling pathways, but in results, discussion or conclusions nothing is said about that.

In the present study, we reported that Cyperus thunbergii extract exhibited vasorelaxant activities by activating endothelial NO synthase. However, we do not demonstrate a direct connection between this vasorelaxant effect and its arginase inhibitory. Thereby, we decide to focus the aim and the discussion of our findings on the arginase inhibitory activities of pure compounds. Vasorelaxant effect of the extract was suppressed. We change the objectives as follows:

Arginase inhibitory activities of polyphenolic enriched extracts were investigated by means of in vitro and ex vivo studies. The isolation, identification and arginase inhibitory activity of seven polyphenolic compounds including two new stilbene oligomers, were also reported here.”

instead of :

“Arginase inhibitory activity and vasorelaxant properties of polyphenolic enriched extracts were investigated, as well as their effects on endothelial function and signalling pathways. Isolation, identification and arginase inhibitory activity of seven polyphenolic compounds including to new stilbene oligomers, were also reported here.”

iii In line 389, page 14 one of the units has a rare symbol, what is this?

The « rare symboles » are typos and have been replaced by the right symbols « µ ».

****

We thank the reviewer for his/her comments that improved the quality of our manuscript.

All the changes made in the manuscript are indicated in yellow in the text (and green for language corrections)

We hope that the revised manuscript comes up to reviewer’s expectations and that the reviewer will consider the corrected version acceptable for publication in Molecules.

Round 2

Reviewer 1 Report

The authors corrected the flaws of the initial version